# Trajectory enhancement of low-earth orbiter thermodynamic retrievals to predict convection: a simulation experiment

Mark T. Richardson[1], Brian H. Kahn[1], Peter Kalmus[1]

[1]Jet Propulsion Laboratory, California Institute of Technology, Pasadena, CA 91109, USA

*Correspondence to*: Mark T. Richardson (markr@jpl.nasa.gov)

**Abstract.** 3-D fields of temperature ($T$) and specific humidity ($q$) retrieved by instruments such as the Atmospheric Infrared Sounder (AIRS) are predictive of convection, but convection often triggers during the multi-hour gaps between satellite overpasses. Here we fill the hours after AIRS overpasses by treating AIRS retrievals as air parcels which are moved adiabatically along Numerical Weather Prediction (NWP) wind trajectories. The approach is tested in a simulation experiment
that samples 3-D European Reanalysis-5 (ERA5) $T$ and $q$ following the real-world AIRS time-space sampling from March—November 2019 over much of the Continental U.S. Our time-resolved product is named ERA5-FCST, in correspondence to the AIRS forecast product we are using it to test, named AIRS-FCST. ERA5-FCST errors may arise since processes such as radiative heating and NWP sub-grid convection are ignored. For bulk atmospheric layers, ERA5-FCST captures 59—94 % of local hourly variation in $T$ and $q$. We then consider the relationship between convective available potential energy (CAPE),
convective inhibition (CIN), and ERA5 precipitation. The 1° latitude-longitude ERA5-FCST grid cells in our highest CAPE and lowest CIN bin are more than 50 times as likely to develop heavy precipitation (>4 mm hr$^{-1}$), compared with the baseline probability from randomly selecting a location. This is a substantial improvement compared with using the original CAPE and CIN values at overpass time. The results support development of similar FCST products for operational atmospheric sounders to provide time-resolved thermodynamics in rapidly changing pre-convective atmospheres.

*Copyright statement.* © 2023 California Institute of Technology. Government sponsorship acknowledged.

## 1   Introduction

Thermodynamic properties such as convective available potential energy (CAPE) and convective inhibition (CIN) are related both to the probability of occurrence and intensity of extreme convective events (Ukkonen and Mäkelä, 2019; Lafore et al., 2017). In-situ radiosonde measurements of the temperature ($T$) and specific humidity ($q$) profiles necessary to calculate such
metrics are spatially and temporally sparse. Satellite retrievals, such as those from hyperspectral infrared (IR) sounders offer improved spatial coverage and have been used to study pre-convective atmospheres (Weisz et al., 2015; Botes et al., 2012; Gartzke et al., 2017). However, convection often triggers hours after the overpass of the low-Earth orbiting (LEO) satellites that host IR instruments, and air motion can greatly change the $T$ and $q$ field between overpass and convective initiation.

We propose accounting for the air motion by using adiabatic parcel theory, assigning parcels an initial $T$ and $q$ at satellite overpass time and then advecting them following NWP winds. However, our method would provide little predictive power if the development of the environment prior to convection is dominated by diabatic processes such as surface heating and small-scale convection. To address these concerns, we perform an idealised simulation experiment using ERA5 as the nature run, and assign initial parcel $T$ and $q$ with ERA5 values from AIRS-sampled locations. Retrieval uncertainty is ignored to evaluate whether air motion alone explains much of the pre-convective development and we find that most ($\geq$50 %) of the variance in local $T$ and $q$ evolution is captured on six-hour timescales. We therefore demonstrate a potential new approach both for nowcasting using current sensors, and for building a multi-decade climate record to study changes in convective conditions.

Our study targets the Central-East-Continental U.S. or CE-CONUS (land within 107—64 °W, 25—53 °N). NOAA (2022) reported increases in "severe storm" (largely convective) >\$1 billion property damage events across the U.S. From 1980—1989 through to 2012—2021, the decadal event count increased from 8 to 85 and inflation-adjusted damages increased from \$12 bn to \$187 bn. Gaps in data and physical understanding of links between anthropogenic warming and storms contribute to disagreement over the precise contribution of increased exposure (e.g., from rising value of exposed properties) versus climate change to reported trends (Barthel and Neumayer, 2012; Hoeppe, 2016).

Hazardous convective weather (HCW) includes thunderstorms, damaging winds, hail and tornadoes and there is currently no consensus on possible climate trends, due to difficulties including event rarity and dataset inhomogeneity (Agee and Childs, 2014). Convection generally occurs in areas identified as favourable for convection from properties such as CAPE, CIN, wind shear and low-level moisture (Tippett et al., 2015), but such conditions do not guarantee that convection occurs. This means that many false warnings may be issued, although convective condition metrics are useful at a statistical level (Brooks et al., 2003). Models generally report increased CONUS convective risk under warming (Trapp et al., 2009, 2007), including some of the latest Coupled Model Intercomparison Project, phase 6 (CMIP) models which show widespread increases in CAPE in particular (Lepore et al., 2021).

In finer-resolution numerical modelling, trends in hail events depend on location, time and intensity (Raupach et al., 2021; Trapp et al., 2019; Mahoney et al., 2012) but a typical finding for non-hail HCW is that weaker events become less common while more intense events become more common. A modelled shift to more intense events has been reported for tornadoes (Krocak and Brooks, 2018) and conditions favourable for convection in general (Rasmussen et al., 2020). A theoretical argument for increases in the most intense convective precipitation is related to the Clausius-Clapeyron (C-C) $\approx$7 % $°C^{-1}$ scaling of saturation specific humidity. There is even limited evidence for a 2×C-C scaling ($\approx$14 % $°C^{-1}$) in some conditions for short-timescale precipitation (Lenderink and Meijgaard, 2008; Lenderink et al., 2017; Busuioc et al., 2016; Westra et al., 2014). Total precipitation change is limited by the atmosphere's ability to lose heat, which is estimated at 1—3 % $°C^{-1}$ (Pendergrass, 2020; DeAngelis et al., 2015), so a 7 % $°C^{-1}$ increase in the most intense precipitation requires less rain falling in weaker events (Trenberth, 2011, 1999; Fischer and Knutti, 2016). Increases in the frequency or intensity of heavy precipitation have now been reported across land areas in observational data or reanalyses (Guerreiro et al., 2018; Chinita et al., 2021; Ali and Mishra, 2018; Donat et al., 2019), including in parts of the U.S.

It is intended that future work will study convective risk over CE-CONUS, and different hazard proxies could be selected (e.g. Heuscher et al. (2022)). The present paper will use ERA5 hourly precipitation as a proxy of convection, since this is directly output by ERA5. Convective proxy datasets with consistent space-time coverage generally have short records, such as from 2014 for the MRMS surface precipitation radar product (Zhang et al., 2016) or since 2017 for geostationary lightning mapping (Rudlosky et al., 2019; Goodman et al., 2013). The short length of proxy datasets motivates the study of thermodynamic trends instead. The Atmospheric Infrared Sounder (AIRS) on NASA's Aqua satellite (Chahine et al., 2006) maintained a Sun-synchronous orbit from 2002-08 until 2021-12. The satellite equator crossing time is now drifting later, but AIRS data is still being processed as of 2022-12 and so provides a unique and valuable multi-decade climate record. For some climate properties, the trend is already sufficiently large relative to internal climate variability to constrain relationships or detect trends, such as for tropical cloud heights using the MODIS instrument on the same satellite platform (Richardson et al., 2022). Our aim is to ultimately exploit AIRS' exceptional instrumental stability (Strow and DeSouza-Machado, 2020) to generate a long-term record of convection-relevant conditions over CE-CONUS.

Reanalyses are otherwise the only other source of multi-decade thermodynamics at hourly resolution, and some earlier studies using them noted a general increase in stability in the inland CONUS and a decrease near the coasts, largely driven by changes in PBL moisture (Trapp et al., 2007; Li and Colle, 2014). More recently, studies using the European Reanalysis 5 (ERA5, Hersbach et al. (2020)) over 1979—2019 showed a complex structure of changes with decreased CAPE, decreased total convective precipitation, and decreased severe storm hours across much of CE-CONUS, along with an increased CIN and 0—6 km wind shear. However, there were often regional differences between ERA5 and local rawinsonde data, raising questions about the fidelity of long-term ERA5 trends in these parameters (Pilguj et al., 2022; Taszarek et al., 2021).

A common issue in reanalyses is that changes in the type of data assimilated result in discontinuities, which then cause trend biases. For ERA5, this has been demonstrated for snow cover (Urraca and Gobron, 2023) and upper-tropospheric temperatures (Shangguan et al., 2019). While ERA5 assimilates AIRS brightness temperatures, using AIRS retrievals alone avoids the reanalysis-specific concerns about changes in assimilated data for $T$ and $q$, offering a complementary perspective on trends in the pre-convective atmosphere. A major limitation of AIRS data is its CE-CONUS overpass time of approximately 13:30 local (18:30 UTC), hours before typical warm-season convective peak (Watters et al., 2021; Kalmus et al., 2019). Our time-resolved product will therefore provide a new source to understand trends in conditions favourable for convection.

A relevant concern with AIRS retrievals has been a historic dry bias near the surface (Botes et al., 2012) resulting in too-stable atmospheres, although this can be addressed by fusing in-situ surface observations (Gartzke et al., 2017; Bloch et al., 2019). Since then, the version 7 AIRS retrievals addressed the near-surface dry bias at the cost of relying more heavily on the retrieval prior and therefore potentially losing relevant local information (Yue et al., 2020).

The AIRS v7 retrieval uses a neural network (NN) to generate its first guess, including $T$ and $q$ profiles. The NN inputs are AIRS measurements, but the NN training dataset was based on ECMWF forecast profiles (Milstein and Blackwell, 2016). The AIRS prior may therefore share common structural biases with reanalysis, but its trends should respond to radiances alone,

since the NN is "expected to behave similarly throughout the whole mission, while model-based first guesses can show significant change in bias structure over mission duration due to model changes" (Yue et al., 2020).

Relevant issues with AIRS retrievals and the potential for climate trend studies will be further discussed in Section 4, but retrieval uncertainties are ignored by design since the purpose of this study is to isolate and quantify errors associated with our method of adding time resolution to AIRS $T$ and $q$ fields. To do so, we take ERA5 $T$ and $q$ fields and convert them to AIRS time and space sampling and then use NWP trajectories to generate a time-resolved product for comparison with ERA5 outputs at later timesteps. We follow AIRS time-space sampling to ensure that our statistical results apply to future work using AIRS data.

The first study to combine NWP trajectories with IR sounder retrievals for convection was Kalmus et al. (2019), who identified certain classifications of HCW events over CE-CONUS, then used the Hybrid Single-Particle Lagrangian Integrated Trajectory (HYSPLIT, Stein et al, (2015)) model to back-trace the parcels to their likely location during the AIRS overpass. Profiles reconstructed from the AIRS-retrieved $T$ and $q$ taken from the back-traced locations were referred to as "trajectory enhanced". Trajectory enhancement generated more realistic pre-convective conditions, such as higher CAPE, than a standard approach of selecting the AIRS profile that was geographically nearest to the later convective event.

The approach was then adapted to go forwards in time and applied to Suomi-NPP and NOAA-20 retrievals from the NOAA-Unique Combined Atmospheric Processing System (NUCAPS). An initial forecast (FCST) version was evaluated during spring 2019 and 2021 at NOAA's Hazardous Weather Testbed (Esmaili et al., 2020). Operationally, NUCAPS-FCST can provide users with information based on the latest satellite $T$ and $q$ fields sooner than if they waited for the next NWP forecast cycle. FCST can also provide complementary information since compared with NWP its results are less sensitive to convective parameterisations.

The FCST codebase has been updated in response to forecaster feedback and its performance is studied in the present paper. The primary motivation is to guide development of AIRS-FCST for climate studies, with a focus on the "FCST" component rather than issues related to any specific set of AIRS retrievals. Nevertheless, a fundamental limitation of any LEO-based product is the spatial sampling of the instrument, so AIRS spatial sampling effects are also studied. Many conclusions should extend to the operational NUCAPS-FCST implemented as part of the Advanced Weather Interactive Processing Systems 2 (AWIPS-II), which uses the same FCST codebase. Here we will specifically evaluate: (i) $T$ and $q$ fields and (ii) whether ERA5 precipitation is indeed more frequent and intense for high CAPE and/or low CIN. Detailed analysis of observational data and of other properties such as wind shear and low-level moisture convergence are left to future work. The results show that ERA5-FCST provides useful predictions of the formation of conditions favourable for convection in ERA5, and supports the development of our upcoming AIRS-FCST product.

This paper is structured as follows: Section 2 describes the data and methodology, Section 3 displays trajectory-enhanced output and reports the performance statistics, then Section 4 discusses and concludes.

## 2    Data and Methods

The AIRS L2 products are provided as granules each containing approximately 6 minutes of data. We first simulate AIRS granules for March—November 2019 by interpolating ERA5 $T$ and $q$ onto the daytime AIRS measurement locations and times over CE-CONUS. HYSPLIT trajectories are then obtained from 2100 UTC of that day to 0200 UTC of the next, which is locally 1500—2000 Central Daylight Time (CDT) for most of the period. $T$ and $q$ values are then averaged within a 3-D grid and CAPE and CIN are calculated from the gridded profiles. The $1°\times1°$ final ERA5-FCST product is then compared with time-matched $1°\times1°$ ERA5 outputs. Supplementary analysis results will be referenced at the appropriate points later, but the overall conclusions hold for a $0.5°\times0.5°$ grid or if using a different NWP data source to drive HYSPLIT.

### 2.1    Construction of ERA5-FCST

For March—November 2019 the hourly ERA5 atmospheric $T$ and $q$ were obtained for all pressure levels with $P \geq 100$ hPa on ERA5's default download $0.25°$ latitude-longitude grid. The 100 hPa limit matches that of the Weather Research and Forecasting (WRF) output used to drive HYSPLIT. ERA5 surface pressure ($P_s$), near-surface temperature ($T_{as}$) and specific humidity ($q_{as}$) were also used. ERA5 is an ideal data source since it provides the necessary fields with horizontal resolution similar to that of AIRS and vertical resolution similar to that of our output ERA5-FCST. This work aims to evaluate the trajectory-enhancement method for adding time resolution to LEO IR products, so we are not concerned about small differences between AIRS and ERA5.

We then selected all AIRS version 7 L2Sup granules that contained infrared-only retrievals with footprints from 1800—2059 UTC within 107—64 °W, 25—52 °N, including over-ocean profiles. For this study's purposes, the only relevant difference between L7 and prior AIRS versions is in the quality control flags. The wide time range captures all AIRS footprints within the domain. ERA5 AIRS-like granules are built as follows: the AIRS time, latitude, longitude and quality flag fields are copied in full, then $P > 100$ hPa levels are copied, leaving 57 of the original 100 levels. ERA5 $T$ and $q$ fields are linearly interpolated onto the AIRS locations in time, then in latitude-longitude, then for the 3-D fields in log-pressure.

HYSPLIT uses Numerical Weather Prediction (NWP) data stored at NOAA's Air Resources Laboratory (ARL). Only two NWP sources, WRF and the NCEP/NCAR reanalysis span the full AIRS time period. We selected WRF for the CE-CONUS analysis because its hourly time resolution and 27 km horizontal resolution are finer than the reanalysis' 6 hours and $2.5°$. A supplementary analysis will be referenced later, and showed that using WRF winds gives better performance than using the NCEP/NCAR reanalysis. The WRF simulations are forecasts for each UTC-defined day (Ngan and Stein, 2017).

Every valid AIRS v7 retrieval with at least "good" quality is used. Spatial sampling therefore matches that of the AIRS v7 product including features such as missing data due to extensive cloud cover. This is an attempt to represent real-world AIRS sampling, the ERA5 cloud cover may differ. Each valid retrieval is treated as a single parcel and its trajectory is calculated with HYSPLIT through 0200 UTC of the next day. To separate retrievals and the trajectory-enhanced parcel values, the

"zeroth" timestep $t_0$ includes 1800—2059 and subsequent values based on HYSPLIT outputs are hourly from 2100 UTC onwards.

HYSPLIT by default requires heights above ground level. AIRS v7 geopotential heights have an indexing issue when the near-surface layer is thinner than 5 hPa so we derive height above ground level using the hypsometric equation with AIRS level $T$, $q$ and $P$, and bottom boundary values from $P_s$, $T_{as}$ and $q_{as}$. Following Kalmus et al., (2019), parcels are then moved to the HYSPLIT calculated location, $T$ is adjusted adiabatically and $q$ is capped to RH≤100 %. Moist adjustments are calculated using SHARPpy version 1.4.0 (Blumberg et al., 2017) and dry adjustments by requiring fixed potential temperature. The mean $T$ and $q$ are then calculated within each 1° latitude-longitude and 30 hPa pressure grid cell, by taking the mean $T$ and $q$ of all parcels falling within the grid cell. CAPE and CIN were calculated with SHARPpy from the gridded profiles for the most-unstable (MU) and mean-mixed-layer (MML) parcels following ECMWF guidance (Groenemeijer and Púcik, 2019). All footprints within 107—64 °W, 25—53 °N are input to HYSPLIT but for the analysis results only 1° grid cells with ERA land fraction >0.5 are included. Past studies used AIRS retrievals at $t_0$, so we will compare ERA5-FCST during 2100—0200 with ERA5-FCST at $t_0$. The 1° gridded ERA5-FCST output at $t_0$ will be labelled ERA5-overpass.

The convective parameters are calculated from the final gridded fields at 30 hPa vertical resolution, and any profile comparisons are based on the same output. We expect our results to be robust to changes in vertical resolution between ERA5, AIRS L2Sup and the final outputs based on a series of resolution sensitivity tests for derived CAPE (Supplementary Figures 1—3, Supplementary Table 1). The AIRS L2Sup vertical layering is also far finer resolution than the "effective" vertical resolution of the retrieval, as discussed in Irion et al. (2018). In reality, the retrieval can only capture smoother changes in profiles than reported on the L2Sup layering, but we also find that our results are likely robust to the AIRS effective vertical resolution (Supplementary Tables 2—3). For FCST the finer L2Sup layering is preferred since it provides more parcels to HYSPLIT.

The only way in which ERA5-FCST parcel $T$ and $q$ can be affected is via vertical motion and the associated adiabatic heating or cooling. We refer to diabatic processes such as radiation, surface fluxes and sub-grid convection as "neglected", even though they indirectly affect results since the NWP simulation that provides the winds for HYSPLIT includes these processes. Nevertheless, the neglected processes can greatly affect $T$ and $q$ profiles in ways that are not captured by FCST. Sub-grid convection in particular can rapidly transport heat and greatly change local profiles, but can still have a relatively small effect on the motion vectors once averaged over a large NWP grid cell. This is because the rising warm air within a grid cell is compensated by nearby descent.

## 2.2  Performance evaluation

The following subsections will detail the methods used to report:

    1)   The representativeness of AIRS sampling during 1800—2059 UTC;

    2)   Skill in forecasting $T$ and $q$ from trajectory enhancement;

3) The predictive skill of the derived CIN and CAPE for predicting ERA5 precipitation.

### 2.2.1 Typical conditions and the representativeness of AIRS sampling

AIRS has nonuniform spatial sampling and a cloud-clearing procedure that permits retrieval of clear-sky properties in the presence of some clouds, but not when cloud cover is too extensive (Susskind et al., 2003). We will compare domain-mean precipitation, CAPE and CIN for CE-CONUS land areas (where land areas are defined as ERA land fraction >0.5) with the means of the complete AIRS swath and of the retrieved profiles alone. The difference between CE-CONUS and the full swath is primarily due to the swath coverage effect, and the difference between full swath and retrievals is primarily due to cloudiness. Cloudiness may indicate past or ongoing convection and correlates with precipitation, so it is important to quantify its effect on mean conditions. The main results will be for the most-unstable parcel MU_CAPE and MU_CIN and results will only be shown for grid cells with valid MU_CAPE estimates.

### 2.2.2 Trajectory enhanced projections of *T* and *q* profiles

Differences in ERA5-overpass relative to ERA5 at $t_0$ result only from the spatial resampling and regridding. The errors introduced by trajectory enhancement can therefore be obtained by comparing $T$ and $q$ from ERA5 during 2100-0200 UTC minus ERA5 at $t_0$ versus ERA5-FCST at each UTC relative to ERA5-overpass. Statistical performance will be reported for changes on the International Satellite Cloud Climatology (ISCCP, Rossow et al. (1991)) layering of "low" ($P$>680 hPa), "mid" (440>$P$>680 hPa) and "high" ($P$<440 hPa). For each layer $i$ and time-step $t$ we calculate $\Delta T_{i,t} = T_{i,t} - T_{i,t_0}$ and $\Delta q_{i,t} = q_{i,t} - q_{i,t_0}$. The ERA5-FCST $\Delta T_{i,t}$ and $\Delta q_{i,t}$ will then be compared with ERA5's, and the Pearson correlation coefficient $r$, root-mean-squared error (RMSE) and bias (ERA5-FCST minus ERA5) will be reported.

If the development of CE-CONUS thermodynamic fields from 2100—0200 UTC is dominated by large-scale motion, then the FCST approach will be more accurate, $r$ will be high and RMSE and bias magnitudes will be small. Other factors will reduce statistical performance. For example, because ERA5-FCST does not explicitly account for surface heating and evaporation, we expect its low levels to be cool and dry relative to ERA5 at later UTC. Meanwhile, due to not accounting for sub-grid convection, we expect that in some cases, ERA5-FCST will have warmer low levels and cooler mid- and upper levels compared with ERA5, since ERA5 will allow effective upwards transport of heat once convection is triggered. The WRF27km runs used in ERA5-FCST include sub-grid convection, but this has little effect on the grid-cell mean winds used in HYSPLIT.

### 2.3.3 Using convective parameters to predict precipitation

ERA5's Integrated Forecast System (IFS) generates both large-scale and convective precipitation (*cp*), but conclusions are not affected by selecting total precipitation *tp* rather than *cp* since *cp* is approximately 89 % of *tp*. This is advantageous since our eventual AIRS-FCST product will be evaluated against real-world *tp* data rather than *cp*. An important caveat is that the ERA5

IFS convection scheme uses CAPE (Bechtold et al., 2014) so the results here may overstate the strength of any derived CAPE-*tp* relationships.

Convective events occur both on days of high and low mean CAPE, so we hypothesised that convection is more likely to occur in areas where local CAPE is high relative to the wider environment. The main text results therefore use what we call local enhancement of CAPE or CIN, which are the grid-cell values minus the daily median of the same property. We label these dMU_CAPE and dMU_CIN. Our convention is to use the absolute value of CIN, such that more positive reported CIN means a more stable atmosphere. Comparisons will also be made with ERA5-calculated CAPE, while ERA5 CIN is ignored since a problem with the ERA5 code means that the ERA5 product values are erroneous (Groenemeijer et al., 2019). The ERA5 "enhanced" CAPE is calculated in the same way as for ERA5-FCST, using the same grid cell sample and subtracting the ERA5 sample's daily median. For simplicity, we refer to this as ERA5 dMU_CAPE since ERA5 product CAPE is derived from its MU parcel (see discussion associated with Supplementary Figures 1—3 and Supplementary Tables 1—3). We bin grid cells according to their percentile of dMU_CAPE and/or dMU_CIN, then within each bin calculate (i) mean *tp* and (ii) frequency with which $tp > 4$ mm hr$^{-1}$, which is the closest round number to the 99.9$^{th}$ percentile of all-hour precipitation, including non-precipitating grid cells.

Using dMU_CAPE as an example, data from all retrieval days are concatenated, resulting in *N* values of dMU_CAPE per UTC hour, where the *N* values have unique combinations of date, latitude and longitude. Thresholds in dMU_CAPE are then calculated from the percentiles of an $N \times 6$ data array, where the $\times 6$ refers to each of the forecast UTC hours. Each location is assigned to a percentile bin, and the associated ERA5 *tp* mean and frequency with which $tp > 4$ mm hr$^{-1}$ are calculated within each bin. This calculation is referred to as "matched time" and represents the performance of ERA5-FCST.

For comparison, we also calculate "overpass time" statistics in which the dMU_CAPE values are simply the *N* ERA5-overpass values of dMU_CAPE repeated for each UTChour. The bin edges for this sample are calculated for the same percentiles, and then the time-varying ERA5 *tp* statistics are calculated for this new bin assignment. In this calculation, each physical location has a single dMU_CAPE value for all six UTChours, but contributes up to six values of *tp* to the calculation. This represents the case of using the same nearest neighbour AIRS sounding for every forecast hour.

The hypothesis justifying the FCST approach is that on relatively short timescales, air parcels conserve some portion of thermodynamic quantities and therefore large-scale motion provides important predictive power for future convective events. We expect that precipitation should be consistently heavier and more frequent in areas of high CAPE and/or low CIN. If high-CAPE and low-CIN conditions in ERA5-FCST are more predictive of precipitation than those conditions in ERA5-overpass, then this is good evidence of the utility of trajectory-enhancement.

# 3   Results

## 3.1   Typical conditions and effect of cloudiness and spatial sampling on regional statistics

AIRS spatial sampling during 1700—2059 UTC over CE-CONUS is shown in Figure 1a. There is almost 100 % coverage over the Midwest and negligible coverage at the domain edges. Figure 1b—c displays mean ERA5 precipitation and CAPE

and their spatial structure implies that AIRS sampling will result in biases in derived area-mean properties.

Figure 2a shows that daily mean precipitation captured by the full AIRS swath is often larger than the CE-CONUS mean, i.e. there is an oversampling of wetter areas by AIRS footprints. As expected, areas with valid retrievals (orange) have lower mean precipitation than the full swath because the associated clouds cause loss of retrievals. During peak summer convection, AIRS' spatial sampling of wetter areas (high bias) is offset by data loss in cloudy areas (low bias). Inspection of daily meteorology

shows that October 2019 saw more spatially extensive storms within the AIRS swath compared with July 2019, and storm-associated data gaps are a major cause of the regional precipitation differences in Figure 2a during October 2019 (Supplementary Figures 4—5). From Figure 2b, MU_CAPE from ERA5-overpass is in good agreement with ERA5 CAPE. Even though convective indices are sensitive to the spatial resolution at which they are calculated and the ERA5 documentation warns of occasional unrealistically high ERA5 values, the strong correlation between ERA5 reported CAPE and that derived

from our profiles using SHARPpy on the ERA5-FCST vertical grid is encouraging. The marginally higher mean CAPE in ERA5-overpass compared with ERA5 in Figure 2(b) may be due to differences in the ERA5 computational approach or in vertical resolution (see text associated with Supplementary Figures 1—3 and Supplementary Tables 1—3). Finally, the ERA5-overpass CIN in Figure 2c shows a weaker annual cycle compared with ERA5-FCST CAPE in Figure 2b.

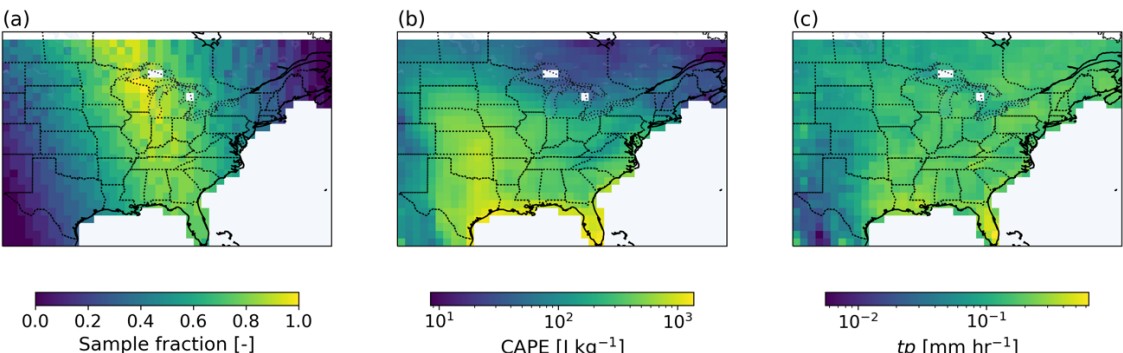

**Figure 1. March—November 2019 means of (a) fraction of days with AIRS swath coverage location, including cloudy-sky footprints, (b) ERA5 mean CAPE during overpass time, (c) ERA5 mean precipitation during overpass time. Grid cells are masked if they contain no land; this includes the ocean and some cells in the Great Lakes.**

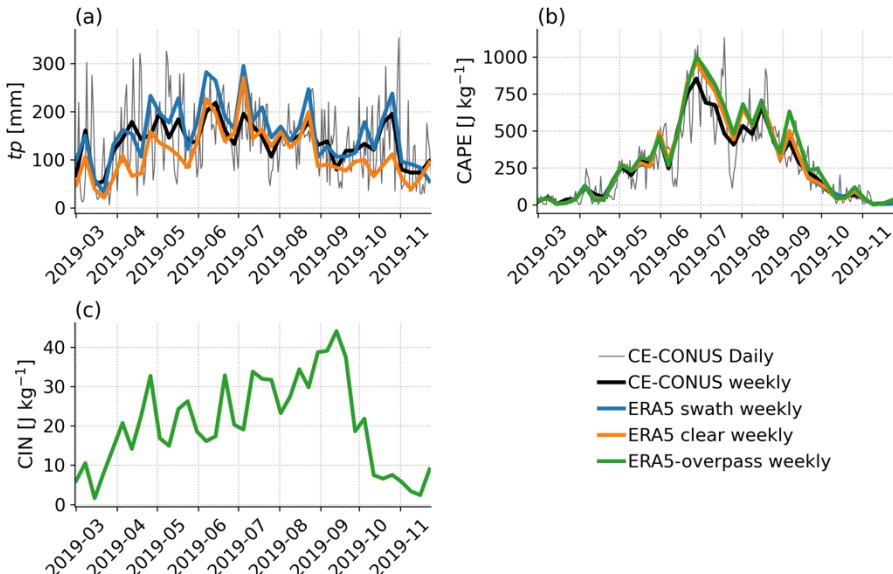

**Figure 2. For land only, area-mean properties during AIRS overpass time $t_0$. In each case the thin black line shows the daily ERA5-reported values for CE-CONUS and the thick black line is the weekly average of the same. Thick blue lines are ERA5-reported values within the full AIRS swath, including non-retrieved footprints. Thick orange lines are ERA5 values averaged over grid-cells with valid AIRS retrievals. Green lines use the same areas as the orange, but as calculated using SHARPpy from ERA5-overpass. (a) precipitation from ERA5, (b) CAPE, (c) CIN. ERA5 CIN values are erroneous and not shown, the ERA5-overpass values are calculated for the most unstable MU parcels.**

Figure 3 displays precipitation and CAPE spatial means for grid cells where ERA5-FCST returns valid CAPE for all UTC of the day. The sampled precipitation peaks in local summer (June-July-August, JJA) and during JJA from 21—22 UTC, several hours after the typical AIRS overpass. The differences in mean CAPE between ERA5 (Figure 3b) and ERA5-FCST (Figure 3c) are consistent with Figure 2b, with generally higher values in ERA5-FCST and the largest discrepancy during September 2019. During JJA the ERA5-FCST CAPE is flat or increasing through 0 UTC, while during the hours of the heaviest precipitation, ERA5's sub-grid convection appears to consume CAPE at a rate greater than it can be generated from daytime near-surface heating and moistening.

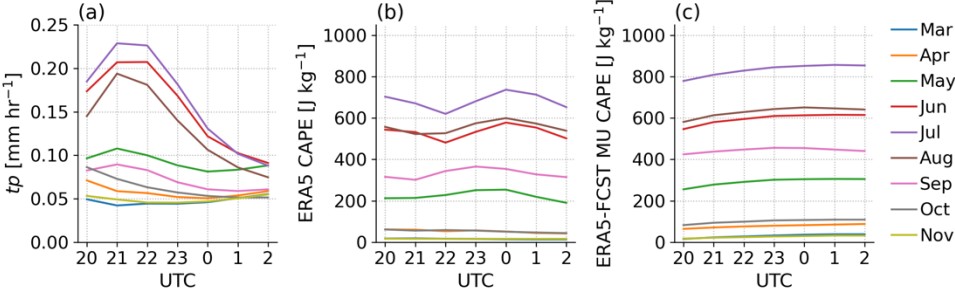

**Figure 3. CE-CONUS hourly (a) ERA5 precipitation, (b) ERA5 CAPE, (c) ERA5-FCST most-unstable CAPE for each calendar month March—November 2019. Only grid cells with valid ERA5-FCST MU CAPE in every hourly timestep are included. This ensures a common spatial map for all hours but will represent different regions than those used in Figure 2 since some grid cells**

with valid CAPE at overpass time will not have valid CAPE at later timesteps. Note that the timestep labelled 20 UTC represents the AIRS sampling time, which varies from 1800—2059 UTC.

## 3.2 Case Study 2019-07-19

During local evening of 19[th] July 2019, the National Weather Service (NWS) reported a band of thunderstorms moving eastwards across Wisconsin, resulting in "damaging winds (60 to 70 mph), several tornadoes, and very heavy rainfall" ([https://www.weather.gov/arx/jul2019](https://www.weather.gov/arx/jul2019) accessed 2022-08-10). Figure 4 shows how the AIRS swath on 2019-07-19 captures the band of high CAPE associated with thunderstorm activity located to the south and west of Wisconsin at overpass time. It also shows that the spatial structure of SHARPpy calculated CAPE (Figure 4b) generally agrees with that calculated by ERA5 (Figure 4a), although the magnitudes differ (see colour bars). West of the band of high CAPE, there is an area of high CIN (Figure 4c). The main results presented henceforth will refer to the 1° resolution results in Figure 4a—c, but 0.5° resolution is shown for comparison in Figure 4d—f. Spatial structures are the same, but cloud-related gaps are larger and striping at the swath edge occurs as the sensor looks off-nadir and the surface footprint locations spread apart.

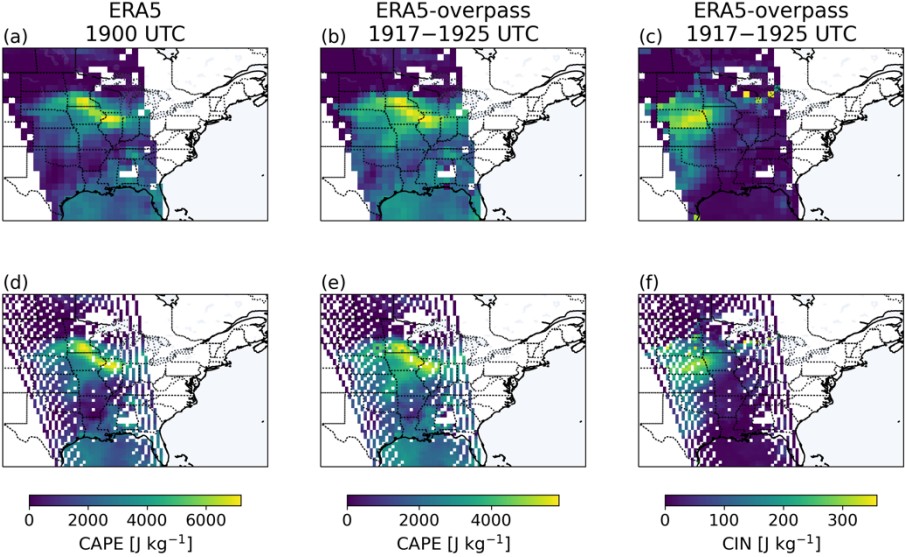

**Figure 4. For 2019-07-19 (a) ERA5 CAPE, (b) ERA5-overpass MU_CAPE, (c) ERA5-overpass MU_CIN. Panels (a—c) at 1° latitude-longitude resolution, (d—f) the same at 0.5 ° latitude-longitude resolution. Note that the colour scales differ between panels, and that ERA5 CIN is not shown due to errors in the product.**

Figure 5(a—f) shows that ERA5-FCST captures the eastward progression of high CAPE and by 0200 UTC, the heaviest precipitation occurred over 44.5 °N, 90.5 °W where ERA5-FCST CAPE rose from 3,300 J kg$^{-1}$ to over 5,400 J kg$^{-1}$. The AIRS overpass could not have indicated the higher risk of convective development by 0200 UTC. ERA5-FCST identified likely storm development here, but there are many areas of high CAPE in Figure 5(a—f) where heavy precipitation does not develop. The results represent the known convective initiation problem: that while heavy precipitation occurred in a high-CAPE areas, high-CAPE did not guarantee heavy precipitation (e.g. Tippett et al. (2015)).

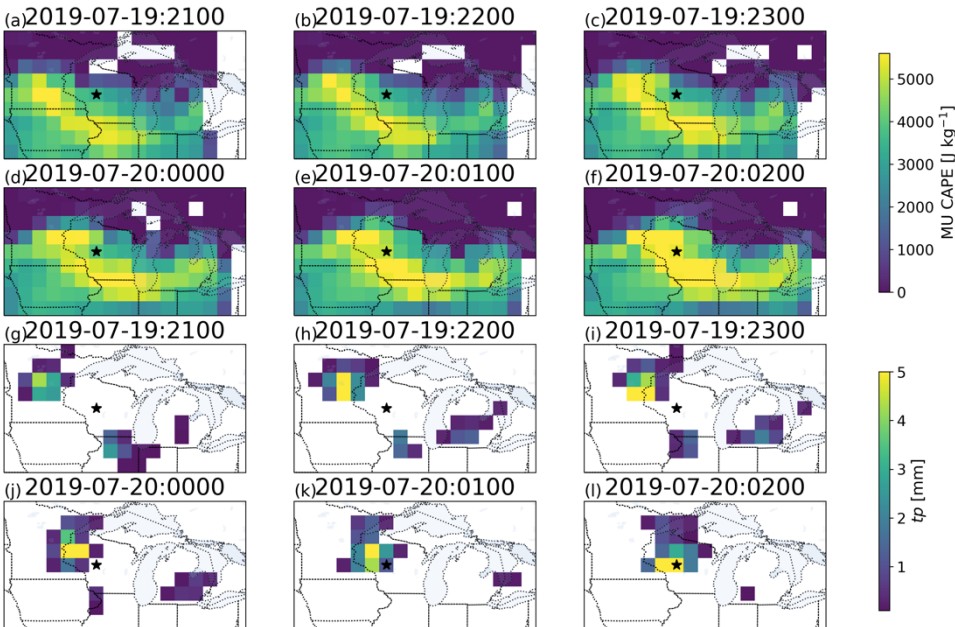

**Figure 5. (a—f) ERA5-FCST calculated most-unstable parcel CAPE over the US upper-Midwest for the six hours starting 2100 UTC. (g—l) ERA5 hourly precipitation in the same area. Star added at 44.5 °N, 90.5 °W represents the location of peak precipitation at hour 6.**

Figure 6 shows the changing $T$ and $q$ profiles at Figure 5's starred location in ERA5 and ERA5-FCST. ERA5-FCST captures important features like the ~800 hPa moistening (bottom row), and the ~100 hPa and ~750 hPa cooling trends (top row). Nevertheless, there are differences such as warming in Figure 6(b) relative to Figure 6(c) from the surface to 800 hPa, likely related to daytime heating which cannot be captured by FCST. As the convection passes overhead after 0000 UTC, it is also notable that the upper troposphere from 100—400 hPa cools substantially more in Figure 6(c) compared with Figure 6(b). The weaker cooling in ERA5 may be explained by sub-grid convection pumping heat into upper levels as the storm passes. Furthermore, Figure 6(e) shows near-surface drying in ERA5 that is not seen in ERA5-FCST (Figure 6(f)). A possible explanation for this is that ERA5 transported near-surface moisture upwards through its sub-grid convection scheme, while the 27 km WRF winds used for ERA5- FCST only include large-scale uplift and so will not capture the related low-level drying; from Figure 5, parcels arriving at the Figure 6 location passed through previously precipitating regions to the west, and precipitation later occurred at the location itself.

The Figure 6 profiles are for a single 1° grid cell that was carefully selected to demonstrate these differences. More representative results are displayed in Figure 7, where changes in temperature (Figure 7a—c) and specific humidity (Figure 7d—f) in ERA5 are compared with those of ERA5-FCST for ISCCP low, middle and high altitude layers. Locations where CAPE is calculated in some but not all timesteps are excluded, ensuring that the spatial sampling is consistent across timesteps.

In this case the time variability in $q$ is better explained than in $T$ with the strongest correlation ($r=0.75$) for the middle layer. For middle and upper layers, the ERA5-FCST product projects larger temperature variation than occurs in ERA5, as was noted for Figure 6(b,c) after convection occurred. This is consistent with sub-grid convection being active in ERA5 but also potentially changes in radiative heating rates dampening $T$ variability. This case study illustrates principles that apply more widely to ERA5-FCST, such as not accounting for surface heating and sub-grid convection, and so help with interpretation of the composite results from the full time period.

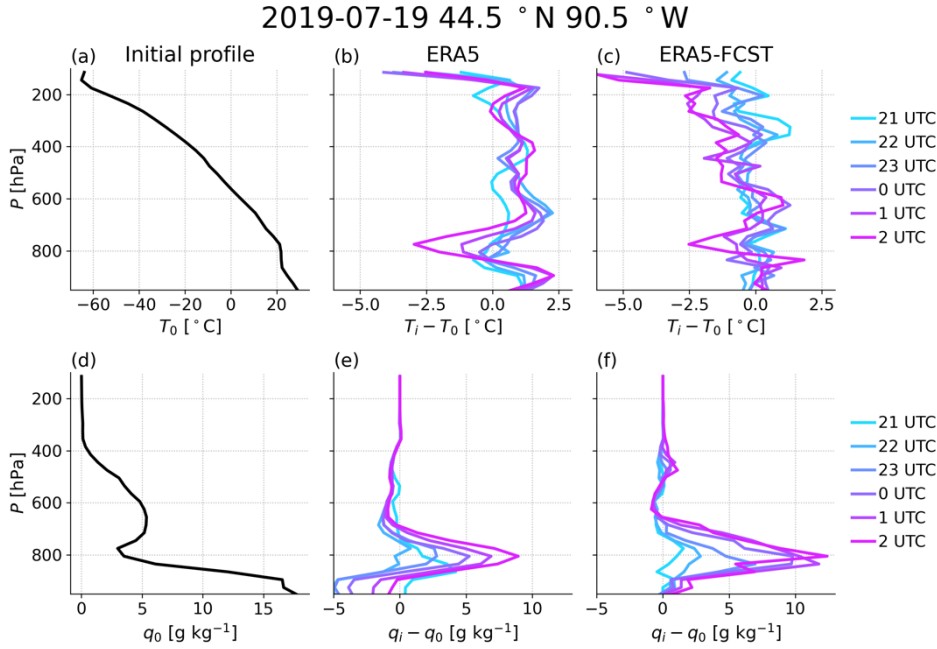

Figure 6. (a) Initial temperature profile $T_0$, (b) difference in ERA5 $T$ relative to $T_0$ during next 6 hours, (c) difference in ERA5-FCST $T$ relative to $T_0$, (d) initial specific humidity profile $q_0$, (e,f) as (b,d) but for moisture. This example was selected to highlight features discussed in the main text, full performance statistics are evaluated later.

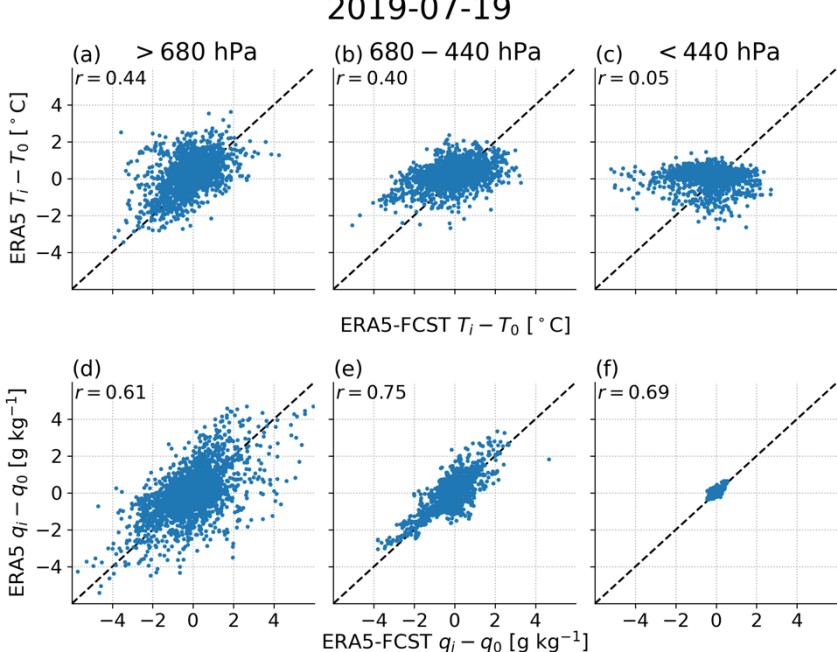

**Figure 7.** (a—c) change in temperature and (d—f) change in specific humidity from 2100—0200 UTC relative to *T* and *q* at overpass time. The *y* axis is the change in ERA5 and the *x* axis is the change in ERA5-FCST. Dashed line is 1:1. Column headed (a) is for *P*> 680 hPa, (b) is for 440>*P*>680 hPa and (c) is for *P*<440 hPa. Each time step is plotted as a single point, so for each layer there are up to six points for each location. Layers correspond to ISCCP low, middle and high and the value in the upper left of each subplot is the Pearson correlation coefficient.

## 3.3  Full period analysis

### 3.3.1    Temperature and specific humidity

The Figure 7 approach is repeated for all March—November 2019 data in Figure 8. Contrary to the 2019-07-19 case, ERA5-FCST generally has better prediction performance for $\Delta T$ than for $\Delta q$. A similarity with the case study is that in Figure 8(b—c) there are regions where ERA5-FCST projects mid- and upper-layer cooling while ERA5 does not. This is consistent with the hypothesis that there are cases where sub-grid convective updrafts occur in ERA5 that are not captured by ERA5-FCST, resulting in warmer air aloft than predicted by the large-scale motion alone. This mid- and upper-layer cooling in ERA5-FCST relative to ERA5 is indeed more common during JJA 2019 when convection peaks (Supplementary Figure 6), or during timesteps when precipitation exceeds the 99[th] percentile (Supplementary Figure 7).

The mean bias reported in each panel of Figure 8 is negative for low- and mid-layer *T* and *q*. The negative *T* difference means that ERA5 shows near-surface warming relative to ERA5-FCST, consistent with the neglect of diurnal heating in ERA5-FCST.

Figure 9 shows that correlations decrease through 21 UTC when convection is more common, before increasing after as large-scale motion once again dominates the spatial structure. The correlation is always $r>0.7$ except for low-level $q$ at 2200 UTC, so despite the neglect of many processes, FCST almost always captures $>50$ % of the time variance in bulk-layer $T$ and $q$ over CE-CONUS following AIRS overpasses. We expect that the unexplained variance is mainly introduced by the neglected diabatic processes, such as sub-grid convection. A further contribution could be from differences between WRF versus ERA5 winds moving parcels in ERA5-FCST to different locations than occurred in ERA5. The relative infrequency of convection ensures that it does not overwhelm the contribution of large-scale motion. The correlation values reported in Figure 8 show that despite the number of diabatic process, which are all neglected, their total contribution to the time evolution of ERA5 layer $T$ and $q$ is smaller than that explained by large-scale motion.

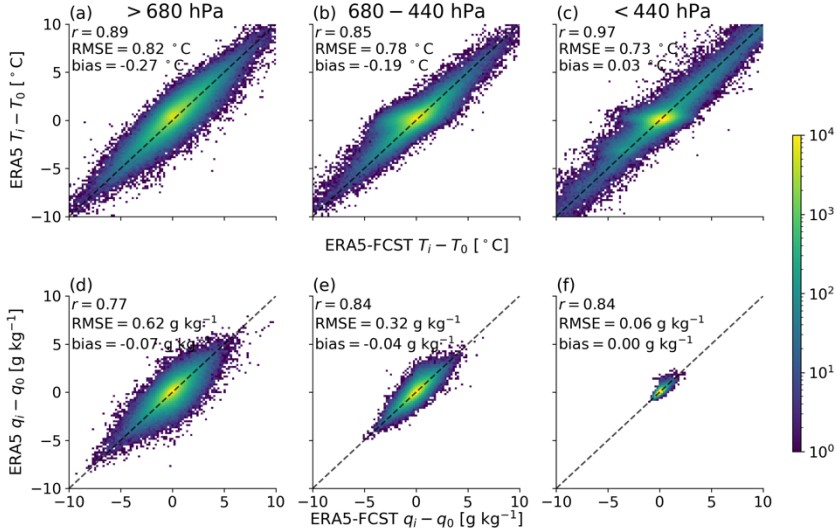

**Figure 8. March—November 2019 relationship between change in (a—c) temperature and (d—f) specific humidity between overpass time and later forecast hours. Dashed line is 1:1. In each case the *x* axis shows the values calculated by trajectory enhancement in ERA5-FCST and the *y* axis contains the actual regridded ERA5 outputs. Columns headed (a) are for low (*P*>680 hPa), (b) for mid (440<*P*<680 hPa) and (c) for high (*P*<440 hPa) layers. Bias is reported as ERA5-FCST minus ERA5.**

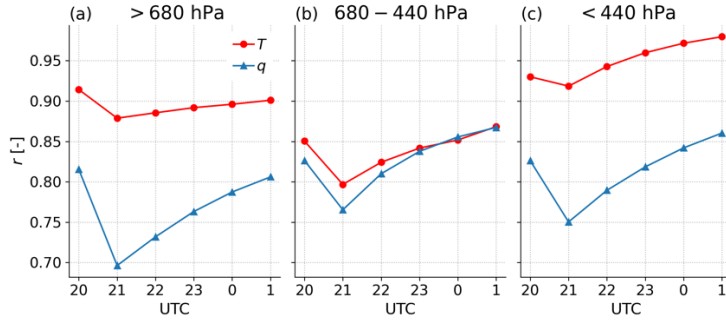

**Figure 9. Change in correlation coefficient between ERA5 and ERA5-FCST for each forecast hour. (a) low levels *P*>680 hPa, (b) mid levels 680>*P*>440 hPa and (c) high levels *P*<440 hPa.**

### 3.3.2 CAPE and CIN individual relationships to precipitation intensity

Figure 10 shows that mean precipitation tends to increase with each dMU_CAPE bin for each hour in each of ERA5, ERA5-FCST and ERA5-overpass. For dMU_CAPE>99.5[th] percentile, there is significantly higher ($p<0.05$) mean precipitation in ERA5-FCST compared with the other CAPE estimates at all timesteps. The higher mean precipitation for high dMU_CAPE represents an improved performance relative to the original AIRS overpass, and this relative improvement gets stronger as forecast hour increases. By forecast hour 4 in Figure 10d, it is only in ERA5-FCST where dMU_CAPE is an obvious predictor of *tp*, and differences relative to other CAPE estimates are significant above the 95[th] percentile of dMU_CAPE. Results are similar for MML parcels and non-enhanced CAPE (not shown). Additional FCST products were processed for JJA only to provide sensitivity tests, including using 0.5° horizontal resolution (Supplementary Figure 8) and non-WRF NWP winds for HYSPLIT (Supplementary Figure 9). Conclusions are qualitatively similar in all cases, but WRF27km has the highest space and time resolution and its product performs best compared with using winds from either the NCEP reanalysis or the Global Data Assimilation System at 1° (GDAS1).

In Figure 10 the ERA5-FCST dMU_CAPE seems more predictive than the ERA5 values extracted from the reanalysis at the same time, which seems surprising. We hypothesise that this is because convective precipitation involves the rapid rise of warm, moist parcels and therefore a strong decrease in local CAPE on sub-hourly timescales in ERA5. When averaged hourly, as in the output used here, precipitating grid cells will be pushed into lower dMU_CAPE bins in Figure 10. ERA5-FCST does not have sub-grid convection, so its CAPE remains high in areas that are likely to precipitate.

Real-world observations using ground-based radiometers support this hypothesis. For example, a mean decrease of 300 J kg[-1] hr[-1] was derived for a set of tornadic storms over Oklahoma (Wagner et al., 2008), and during a study on lightning initiation in China, CAPE decreases during storms sometimes exceeded 40 J kg[-1] minute[-1] (Pan et al., 2020). In these convective events, hour averaging therefore reduces the predictive power of CAPE.

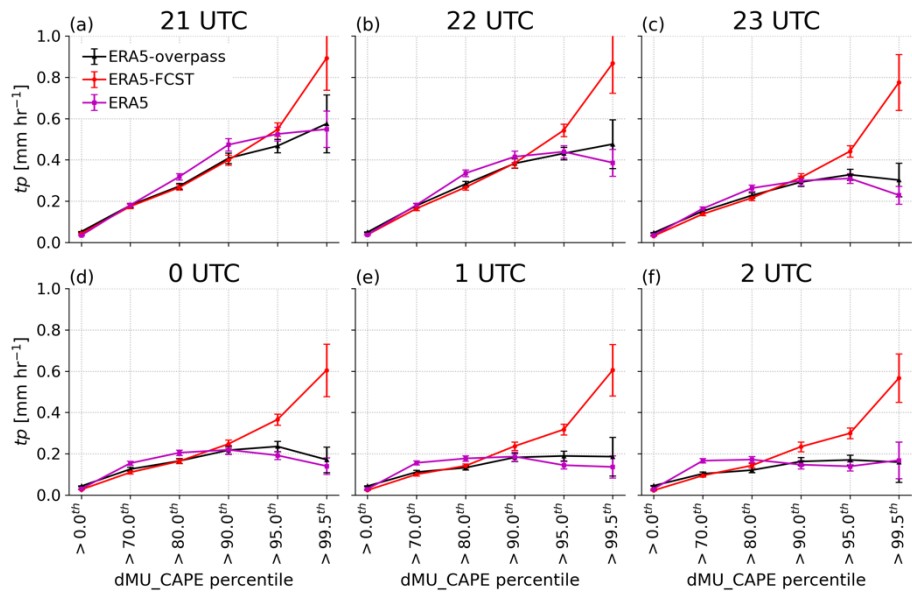

**Figure 10. Mean precipitation ±2σ standard error for each forecast timestep, binned by percentile in MU CAPE (ERA5-overpass in black or ERA5-FCST in red) or by ERA5 surface product CAPE (magenta).**

For CIN, ERA5 is excluded due to its output errors but the ERA5-FCST behaviour is somewhat different than that of CAPE since 62 % of profiles have most-unstable parcels with CIN=0 J kg$^{-1}$. In Figure 11 mean $tp$ peaks for the 70—80$^{th}$ percentile of dMU_CIN, and the behaviour is similar for MU_CIN (not shown). The smallest dMU_CIN bin mostly consists of parcels whose actual MU_CIN = 0 J kg$^{-1}$ and also with low mean actual MU_CAPE (~200 J kg$^{-1}$). The 70—80$^{th}$ percentile dMU_CIN bin generally includes grid cells with absolute CIN of up to approximately 25 J kg$^{-1}$, these grid cells also have the highest mean MU_CAPE (~1,100 J kg$^{-1}$), which likely explains this bin's $tp$ peak. Effectively, there are large areas of low CAPE and low CIN where precipitation does not occur, and then in many cases with large CAPE there is also a small amount of CIN. A fraction of these high-CAPE parcels convect, and result in more average precipitation than the lowest-CIN and lowest-CAPE regime. When we subtract the daily median to obtain our enhanced dMU_CIN values, the 70—80$^{th}$ percentile bin still contains parcels with the highest mean CAPE.Above the 80$^{th}$ percentile of dMU_CIN there is decreasing mean $tp$ as inhibition strengthens. From 22 UTC onwards, higher dMU_CIN percentiles in ERA5-FCST show significantly ($p<0.05$) suppressed $tp$ compared with ERA5-FCST at $t_0$ except for the highest bin capturing the >99.5$^{th}$ percentile.

Figure 10 and Figure 11 are consistent with the FCST procedure improving the predictability of precipitation relative to AIRS overpass, in that high dMU_CAPE coincides with higher mean precipitation, while high dMU_CIN coincides with lower mean precipitation. The figures also demonstrate that predictability depends on the forecast hour, with highest mean $tp$ at earlier UTC, while the difference between ERA5-overpass and ERA5-FCST grows with forecast hour. In each figure, the larger the difference between the black and red lines, the larger the effect introduced by our trajectory-enhancement procedure. The

interpretation of the CAPE or CIN results individually is complicated by their cross-correlation, so the next step is to consider the joint effect of CAPE and CIN.

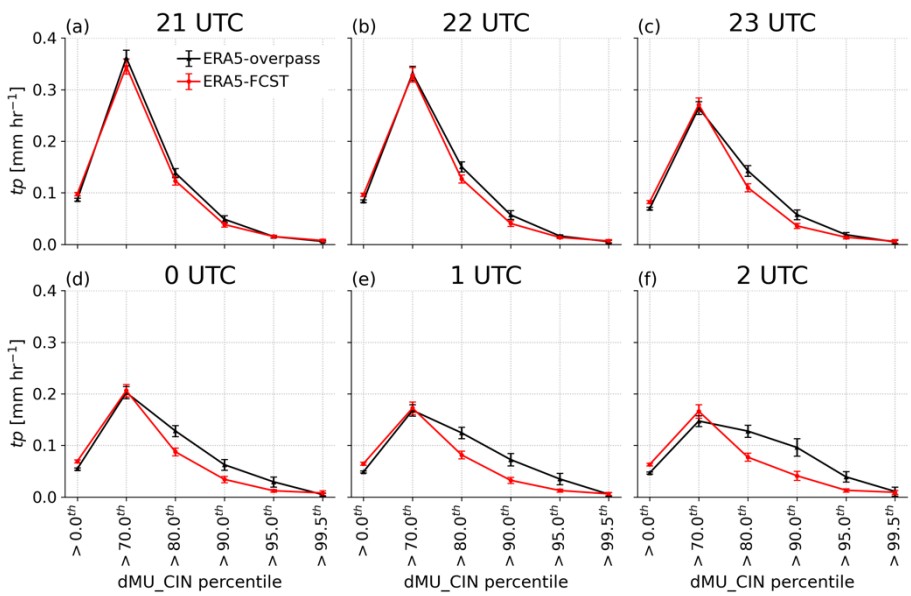

**Figure 11. Mean precipitation ±2σ standard error in each 1°×1° grid cell binned by dMU_CIN percentile, panels (a)—(f) cover 20 UTC of observation day to 1 UTC of the next day as titled. Our convention is to use absolute CIN, such that positive values (higher percentiles) represent more stable profiles. ERA5-overpass (black) means using the CIN calculated for the most-unstable (MU) parcel from timestep 0, i.e. the profiles sampled at AIRS overpass time for that location. ERA5-FCST (red) uses MU CIN calculated**
**from the trajectory-enhanced profiles at the same time as well as location.**

### 3.3.3 Combined effect of CAPE and CIN on precipitation intensity and frequency

Figure 12a,b shows that mean precipitation is indeed highest for high dMU_CAPE and weak dMU_CIN. As dMU_CIN increases from left to right, the corresponding *tp* decreases as expected. Most importantly, the relationship is substantially stronger in Figure 12b than in Figure 12a, representing improved predictive ability when using the time-matched ERA5-FCST
output. For the ERA5-FCST minus ERA5-overpass differences in Figure 12c,d, improvements are represented by red in the upper left and blue in the lower right of each panel. The results show substantially improved performance for ERA5-FCST, with the highest dMU_CAPE plus lowest dMU_CIN bin seeing a +160 % enhancement in mean precipitation relative to using the overpass-derived convective parameters.

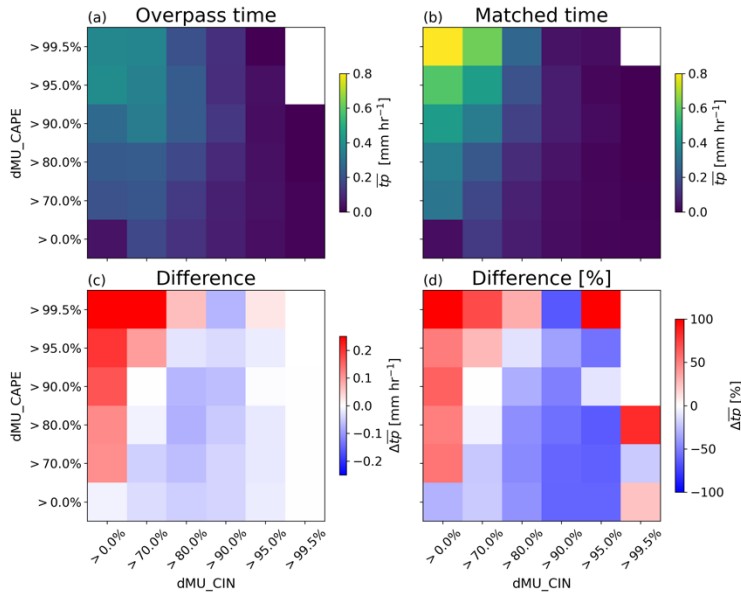

**Figure 12. (a) mean precipitation in each dMU_CIN and dMU_CAPE percentile bin using the ERA5-overpass values derived from overpass time. (b) the same but for time-matched ERA5-FCST values. (c) the absolute difference in mean precipitation, i.e. ERA5-FCST minus ERA5-overpass. (d) the same but expressed as a percentage of the ERA5-overpass values. See Sec. 2.3.3 for the definition of overpass time versus matched time calculations.**

For frequency with which $tp$ exceeds 4 mm hr$^{-1}$ the results in Figure 13 are starker. In the full sample, $tp>4$ mm hr$^{-1}$ in 0.13 % of all timesteps, so if a set of locations were randomly selected then an average of 0.13 % of those locations would have $tp>4$ mm hr$^{-1}$. Any relationships that reliably predict a set of locations where frequency deviates substantially from 0.13 % therefore provide information about when such precipitation is likely to occur. Once again, this happens for the combination of high dMU_CAPE and low dMU_CIN, even if the ERA5-overpass results are used on their own. The top-left bin in Figure 13a

represents a probability of occurrence of 0.9 %, or an enhancement by a factor of 7 in the probability of occurrence. The ERA5-FCST results in Figure 13b show a large improvement, with the upper-left bin frequency being 6.4 % or a factor of 50 increase in probability of heavy precipitation. Once again, the Figure 13c,d differences have the distinctive gradient of red in the upper left and blue in the lower right that are characteristic of improved predictability when using ERA5-FCST rather than ERA5-overpass. Results for both intensity and frequency are similar regardless of the CAPE and CIN definitions, or when using

thresholds other than $tp>4$ mm hr$^{-1}$ to define heavy precipitation (Supplementary Table 4). Overall, the presented results show strong evidence that trajectory enhancement (i.e., our "FCST" method) improves predictability of the mean intensity of precipitation and the frequency of heavy precipitation, compared with satellite data from overpass time alone.

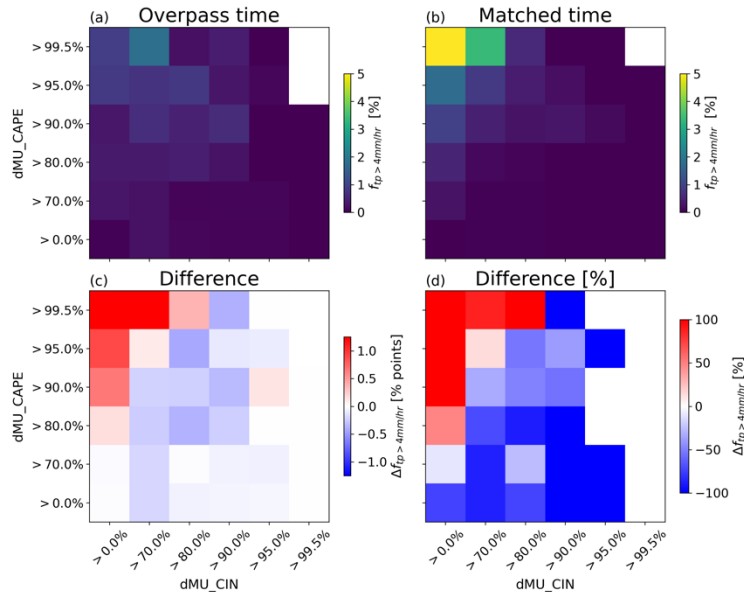

**Figure 13. (a) frequency with which precipitation exceeds 4 mm hr⁻¹ in each dMU_CIN and dMU_CAPE percentile bin using the ERA5-overpass values derived from overpass time. (b) the same but for time-matched ERA5-FCST values. (c) the absolute difference in precipitation frequency, i.e. ERA5-FCST minus ERA5-overpass in percentage points. (d) the same but expressed as a percentage of the ERA5-overpass frequency. See Sec. 2.3.3 for the definition of overpass time versus matched time calculations.**

## 4      Discussion

This paper presents evidence from a simulation experiment that trajectory enhancement can fill time gaps in thermodynamic fields between overpasses by low-Earth orbiting sounders such as AIRS. The approach here builds on Kalmus et al. (2019) by generating forward trajectories from AIRS retrievals rather than backwards trajectories from known convective events. The code is being used operationally with NUCAPS retrievals for nowcasting applications, and for climate studies, an AIRS-FCST product is in production. It will then be possible to identify trends in conditions favourable for convection across CE-CONUS using a record built from a single, highly-stable instrument.

For AIRS, the afternoon CE-CONUS swaths preferentially sample the wetter part of the domain (Figs. 1 and 2), and this must be considered in comparing AIRS-derived changes with model expectations. Fortunately, clouds do not appear to introduce a large-scale sampling bias to CAPE (Fig. 2). Precipitation peaks hours after AIRS overpass, temporarily suppressing CAPE in ERA5 but not ERA5-FCST (Fig. 3), likely due to sub-grid convection in ERA5. The large-scale WRF winds used in HYSPLIT could include some vertical motion carried over from its own convection scheme, but it clearly does not capture the smaller-scale plumes that can locally be responsible for substantial vertical heat transport.

Using the case study of severe storms over Wisconsin on 2019-07-19, Figs. 4 and 5 showed that a FCST approach can accurately capture the progression of conditions favourable for convection, including when CAPE development is driven by large-scale moisture convergence (Fig. 6). When severe weather occurs, mid- and upper-layer changes in $q$ are more

predictable than those of $T$ (Fig. 7). For the 9 months of data including non-severe weather days, $T$ is marginally more predictable than $q$ and >50 % of the temporal variance for each of the ISCCP vertical layers is captured by ERA5-FCST (Fig. 8). The largest source of discrepancy is where ERA5 mid- and upper-layer $T$ is warmer than projected by ERA5-FCST. Factors such as radiation could play a role, but the differences peak in JJA and during the hours of peak convection (Fig. 9), suggesting that sub-grid convection is the largest factor causing errors in ERA5-FCST. All diabatic processes are ignored, and grid cells may include air that was transported from a location that was not sampled by AIRS. Despite these limitations, most of the time variation in bulk-layer $T$ and $q$ is captured by FCST's implementation of adiabatic parcel theory and only accounting for large-scale motion.

Finally, ERA5 precipitation is predictably more intense or frequent when ERA5-FCST reports higher CAPE or lower CIN, compared with predictions made using the overpass time values, both individually (Figs. 10, 11) or combined (Figs. 12, 13). We used ERA5 precipitation for consistency and as a proof of concept, but the relationship between convective parameters and precipitation may differ between ERA5 and the real world, which will be the target of future study.

It may be surprising that ERA5-FCST CAPE showed greater predictive ability than using ERA5 CAPE directly, but when convection occurs in ERA5 it will consume CAPE and result in a lower hourly-mean value in the outputs. In this case, the lack of sub-grid convection in ERA5-FCST means that its CAPE will be more sensitive to large-scale moisture convergence and this may explain why it is more predictive of heavier mean precipitation. There is a downside in that if precipitation does occur, then subsequent FCST timesteps will have artificially high CAPE, and so it may lead to more false alarms in later hours. Despite the simplification of using only CAPE and CIN as predictors for precipitation, ERA5-FCST shows strong improvement in predictive skill relative to ERA5-overpass alone. In particular, the likelihood of precipitation of >4 mm hr$^{-1}$ intensity in its high-CAPE/low-CIN bin is 6.4 % compared with the naïve climatology of 0.13 % or the daily overpass value of 0.9 %. Its high CIN/low CAPE bins have such precipitation occur in ≤0.006 % of cases. These are promising improvements in predictive ability, but even an increase in probability to 6.4 % means that heavy precipitation did not occur locally in the ERA5 reanalysis in the vast majority of the highest CAPE/lowest CIN cases.

The present study has tested the FCST approach to generating sub-daily time resolved fields from LEO IR sounder data, and given the positive results we are processing an AIRS-FCST based on the v7 IR-only retrievals. Future work will involve studying the consequences of uncertainty introduced by the use of real AIRS retrievals, and in interpreting the relationship to real-world $tp$ using some of the understanding developed through this idealised ERA5-FCST study. For example, we may find improved prediction after accounting for the spatial distribution and time development of convection-relevant properties such as wind shear and low-level moisture or moisture convergence. The relationship between FCST outputs and precipitation may also depend on the time of year or day, or perhaps on location within the AIRS swath. It is less likely that air parcels from outside the swath will, at later forecast hours, advect into the centre of the swath compared with the edges.The results here are further tempered by the restriction to precipitation as a convective proxy, which was selected due to the availability of ERA5 outputs and because multi-year, spatially complete MRMS data will be used in future real-world studies. This does mean that specific risks associated with tornadoes or hail, for example, are ignored. Furthermore, AIRS retrieval errors were ignored and

these may degrade the ability to predict convective risks, particularly due to its lower sensitivity to boundary layer moisture. Earlier work noted a bias in AIRS low-level moisture, which has partially been addressed in Version 7 at the cost of losing

sensitivity to the measurements in favour of the prior assumed state (Yue et al., 2020).

Despite Yue et al. (2020)'s expectations that the AIRS neural network (NN) prior is unlikely to cause trend biases, machine learning algorithms can drift if the environment differs from their training set. Prior-related biases in any AIRS-derived product should therefore be independent of reanalysis trend biases, since ERA5 assimilates brightness temperature rather than AIRS retrievals. Therefore, AIRS-FCST will provide independent information to study climate trends. Furthermore, future AIRS

product development could use an updated training set to remove environmental drift. Ongoing AIRS-FCST development work will investigate AIRS-related issues in more detail, while this paper has established a deeper understanding of FCST related issues.

The 2002—recent AIRS-FCST record of thermodynamics will be used with the MRMS surface radar (2014—recent) to relate the derived thermodynamics to convection. In a Bayesian sense, AIRS-FCST will provide $P$(thermodynamics) and to obtain

our target of $P$(convection) we aim to derive $P$(convection|thermodynamics) using the combination of AIRS-FCST and MRMS. The proposed analysis is subtly different from previous work such as Kalmus et al. (2019), which studied thermodynamics in convective versus non-convective atmospheres and so reported results relevant to the inverse problem of $P$(thermodynamics|convection). We also emphasise that while the present study considered CAPE and CIN, this is a proof of concept that only considered a subset of potential thermodynamic properties.

Following on from Kalmus et al. (2019), this is the first paper to systematically study the performance of trajectory enhancement as a method to fill time gaps between satellite overpasses. Prior work was limited by its data sources to specific events or subsets of cases, and by limited validation data. Here we show that trajectory enhancement provides a consistent and promising improvement in the ability to capture changes in the pre-convective environment. Trajectory enhancement could become a useful method for the study of the pre-convective environment, complementing other forecast tools or providing

coverage in regions that lack regular meteorological data. We are currently developing an AIRS-FCST record and will be guided by this simulation study in assessing its strengths and limitations. Successful production and validation of an AIRS-FCST record would allow a detailed study of changes in thermodynamic-driven convective risk over CE-CONUS over the past two decades.

**Acknowledgements:**

Research was carried out at the Jet Propulsion Laboratory, California Institute of Technology, under a contract with the National Aeronautics and Space Administration (80NM0018D0004). Financial and in-kind support for this project was provided by NASA ROSES Science of Terra, Aqua and Suomi-NPP program. A portion of this work was funded through the Joint Polar Satellite System (JPSS) Proving Ground and Risk Reduction (PGRR) program by the National Oceanographic and

Atmospheric Administration (NOAA). We thank Dr. Ivan Tsonevky at ECMWF and Dr. Peter Groenemeijer at the European Severe Storms Laboratory for assistance regarding ERA5 convective parameters, and Dr. Evan Fishbein at JPL for helpful

discussion regarding AIRS retrievals and information content. The contents in this manuscript are solely the opinions of the authors and do not constitute a statement of policy, decision or position on behalf of NASA, the Jet Propulsion Laboratory, or the US Government. High Performance Computing resources used in this investigation were provided by funding from the JPL Information and Technology Solutions Directorate. Copyright 2023. All rights reserved.

**Data and code availability:** The ERA5-FCST data and collocated ERA5 data used in the main analysis are stored at the JPL open repository at https://dataverse.jpl.nasa.gov/dataset.xhtml?persistentId=doi:10.48577/jpl.EESTWM (free access). ERA-5 data is available from the Copernicus Climate Data Store at https://cds.climate.copernicus.eu/#!/home (free, registration required), HYSPLIT is available from NOAA at https://www.ready.noaa.gov/HYSPLIT.php (free, registration required) and AIRS v7 L2 granules are available from the Goddard Earth Sciences Data & Information Services Center (GES DISC) at https://disc.gsfc.nasa.gov/datasets/AIRS2SUP_7.0/summary (free, registration required). SHARPpy is on github at https://github.com/sharppy/SHARPpy or may be downloaded using conda's conda-forge channel (free). The WRF27km dataset was downloaded from the NOAA ARL archives https://www.ready.noaa.gov/archives.php (free).

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
