# Peer review of "Trajectory enhancement of low-earth orbiter thermodynamic retrievals to predict convection: a simulation experiment"

_EGUsphere, 2023_

## Author Comment (AC1)

**REVIEWER 1**

**This paper assesses the usefulness of satellite-measured atmospheric temperature and humidity for convection prediction. The assessment is based on surrogate data as opposed to real measurements. Specifically, global reanalysis data are sampled according to the sampling pattern of a polar orbiting satellite, to mimic the retrievals of an infrared hyperspectral sensor, AIRS. One interesting aspect of this investigation is the use of a trajectory model, which was introduced in an earlier work (Kalmus 2019), to increase the spatiotemporal representativeness of the satellite measurements. The paper is logically organized and well written, providing sufficient technical information and clear descriptions of the results. I do have some concerns, as detailed below, on several aspects of the paper, including the design of the research, the method, and the interpretation of some results. I think this paper could add excellent contribution to the literature after these comments are addressed.**

Thanks for carefully reviewing our paper and for your encouraging comments.

We respond point-by-point below but first summarise two points that merit particular attention.

Firstly, we provided more context and justification for our experimental design choices, in particular why we include all valid profiles even after convection has happened. Our planned uses involve predicting convection from thermodynamics. Predicting "does" versus "does not" convect cannot be reliably done by only learning from cases in which it does. In a new discussion paragraph, we describe our plans and use Bayesian framing to explain our choices. The argument is specific to our purposes though, we did follow one of your suggestions testing $T$ and $q$ statistics during precipitation. The results are shown in a new supplementary figure and are useful for testing some of our statements regarding processes.

Secondly you raise important issues related to sounder retrievals. We want to evaluate FCST without restricting results to a single sounder product. We've added text but tried to ensure that conclusions will not *only* apply to a particular AIRS product version. New discussion covers AIRS v7's non-analysis priors and potential trend biases. Then a new supplementary analysis studies the consequences of changes in vertical & horizontal resolution for our conclusions. The new analysis is general in its treatment of resolution, so that conclusions should be easily translated to other potential sounder products, but we briefly report results from actual AIRS data in the supplement too.

We hope that the additional text and supplementary analysis satisfies your concerns and once again, we appreciate your very thoughtful commentary.

Your review text is **bold black**, our commentary is magenta (this colour!) and any quoted text that is in the main paper is in quotation marks and is "dark green".

**L30-35. Two points are provided as the motivation of this work: weather and climatology. These starting points probably need to be reflected on or revised. For the objective of improving weather forecast, since the trajectory relies on NWP-model modelled winds, how could this approach have any advantage over the data assimilation approach? For the objective of studying convection climatology, why not simply use the reanalysis data without reducing the sampling to match AIRS?**

We have changed text to better justify. AIRS-FCST aims to allow study of climate trends with some independence from other methods. This is analogous to how e.g. trends in SSTs have been studied with buoys, ships, Argo floats & satellite products to increase our overall confidence in the real physical changes, even if each approach has its own uncertainties.

The new text includes:

"A common issue in reanalyses is that changes in the type of data assimilated result in discontinuities, which then cause trend biases. For ERA5, this has been demonstrated for snow cover (Urraca and Gobron, 2023) and upper-tropospheric temperatures (Shangguan et al., 2019). While ERA5 assimilates AIRS brightness temperatures, using AIRS retrievals alone avoids the reanalysis-specific concerns about changes in assimilated data for T and q, offering a complementary perspective on trends in the pre-convective atmosphere."

Using a single data source avoids one type of common issue that results in trend biases. All LEO sounders have sampling issues, so we need to include that factor to address our research goals:

"we take ERA5 T and q fields and convert them to AIRS time and space sampling and then use NWP trajectories to generate a time-resolved product for comparison with ERA5 outputs at later timesteps. We follow AIRS time-space sampling to ensure that our statistical results apply to future work using AIRS data."

And repeat the point for emphasis at the end:

"The primary motivation is to guide development of AIRS-FCST for climate studies, with a focus on the "FCST" component rather than issues related to any specific set of AIRS retrievals. Nevertheless, a fundamental limitation of any LEO-based product is the spatial sampling of the instrument, so AIRS spatial sampling effects are also studied."

With reference to NUCAPS, we mention some advantages,:

"Operationally, NUCAPS-FCST can provide users with information based on the latest satellite T and q fields sooner than if they waited for the next NWP forecast cycle. FCST can also provide complementary information since compared with NWP its results are less sensitive to convective parameterisations."

Based on present tests we expect to be able to get the NUCAPS-FCST fields into AWIPS II within ~1 hour of overpass.

**L82. An important claim is made here about AIRS being advantageous for studying climate trends compared to reanalyses. This point needs to be better discussed, as one can easily come up with counterarguments. For example, given that the conventional retrievals typically take prior information including first guesses from analysis, it is not obvious to me that the retrieval products aren't subject to the same issues as reanalyses. A general comment is that I think the paper can provide better reasoning or more references to establish suitability of AIRS for studying climate trends. For instance, do you think the radiometric stability of AIRS together with its spectral information may facilitate detecting convection regime changes, taking advantage of their spectral signatures (e.g., Huang and Ramaswamy 2008, https://doi.org/10.1029/2008GL034859; Kahn et al. 2016, https://doi.org/10.1002/2016GL070263)? Or, may methods particularly designed for climate trending, such as the average-then-retrieve approach (e.g., Huang et al. 2010, https://doi.org/10.1029/2009JD012766; Kato et al. 2014, https://doi.org/10.1175/JCLI-D-13-00566.1) be of relevance here?**

We agree with the reviewer that more finesse was needed in the supporting argument. We have added the following text to the introduction, which covers how the v7 retrieval first guesses use only the radiances as input:

"The AIRS v7 retrieval uses a neural network (NN) to generate its first guess, including T and q profiles. The NN inputs are AIRS measurements, but the NN training dataset was based on ECMWF forecast profiles (Milstein and Blackwell, 2016). The AIRS prior may therefore share common structural biases with reanalysis, but its trends should respond to radiances alone, since the NN is "expected to behave similarly throughout the whole mission, while model-based first guesses can show significant change in bias structure over mission duration due to model changes" (Yue et al., 2020).

Relevant issues with AIRS retrievals and the potential for climate trend studies will be further discussed in Section 4, but retrieval uncertainties are ignored by design since the purpose of this study is to isolate and quantify errors associated with our method of adding time resolution to AIRS T and q fields."

Then in Section 4 we further discuss:

"Despite Yue et al. (2020)'s expectations that the AIRS neural network (NN) prior is unlikely to cause trend biases, machine learning algorithms can drift if the environment differs from their training set. Prior-related biases in any AIRS-derived product should therefore be independent of reanalysis trend biases, since ERA5 assimilates brightness temperature rather than AIRS retrievals. Therefore, AIRS-FCST will provide independent information to study climate trends. Furthermore, future AIRS product development could use an updated training set to remove environmental drift. Ongoing AIRS-FCST development work will investigate AIRS-related issues in more detail, while this paper has established a deeper understanding of FCST related issues."

We considered discussion about AIRS product trends in general, but didn't see an obvious place to enter those citations. With the thermodynamics we are targeting hourly dynamics, and the rest of the discussion touches on hazardous convective weather, which is not well-captured at AIRS overpass time. We think our work is substantially different from the citations you mention, author Kahn supports the choice to exclude these citations.

**L129. A critical methodological question here is whether ERA5 profiles can appropriately represent the vertical resolution of AIRS retrieval or its ability in measuring such quantities as CAPE. I'm surprised that this important consideration is completely neglected. The paper would benefit from a proper discussion of this issue or an assessment of the impacts, for instance, by using the AIRS averaging kernels.**

This is a very good point and we should provide readers with evidence. Rather than substantially lengthening the paper, we have added a supplementary analysis and refer to it in the main text via:

"The convective parameters are calculated from the final gridded fields at 30 hPa vertical resolution, and any profile comparisons are based on the same output. We expect our results to be robust to changes in vertical resolution between ERA5, AIRS L2Sup and the final outputs based on a series of resolution sensitivity tests for derived CAPE (Supplementary Figures 1—3, Supplementary Table 1)."

The supplementary analysis prefers CAPE-calculation sensitivity tests because we believe it sufficiently addresses the point of spatial resolution. An averaging kernel analysis would have been lengthy and difficult to explain to readers who're unfamiliar with standard Bayesian retrieval frameworks.

We show how our results aren't very sensitive to vertical or horizontal averaging, with figures such as a new Supplementary Figure 2, reproduced here:

[Figure]

ERA5 CAPE represents the result using the highest possible resolution, including 137 vertical levels. Associated supplementary text then goes into more detail to show how our *binning* of grid cells by CAPE means that our results are not sensitive to nonlinearities or mean differences in CAPE between different calculation methods. It is only if the relationship is not monotonic or if scatter is large that our results are sensitive to CAPE calculation method and vertical resolution.

The paper tries to distinguish between issues related to retrievals (AIRS *or* NUCAPS, although we mainly discuss AIRS) and the FCST procedure. We simply have to establish whether FCST is at all valid before we can justify any products using this technique, and we believe the present submission quantifies its performance in a suitable manner.

The consequences of retrieval errors will be investigated as we progress, but we report some preliminary statistics based on resampling AIRS v7 L2Sup onto the ERA5-FCST grid (1° horizontal, 30 hPa vertical) at the end of the new supplementary analysis.

**L155/L319. What "neglected processes" are referred to here?**

Where introduced (paragraph near original L155) we are now more explicit:

"The only way in which ERA5-FCST parcel T and q can be affected is via vertical motion and the associated adiabatic heating or cooling. We refer to diabatic processes such as radiation, surface fluxes and sub-grid convection as "neglected", even though they indirectly affect results since the NWP simulation that provides the winds for HYSPLIT includes these processes. Nevertheless, the neglected processes can greatly affect T and q profiles in ways that are not captured by FCST. Sub-grid convection in particular can rapidly transport heat and greatly change local profiles, but can still have a relatively small effect on the motion vectors once averaged over a large NWP grid cell. This is because the rising warm air within a grid cell is compensated by nearby descent."

**L205. A relevant question of interest is how much the 1:30am/pm overpass times of AIRS limit the convection prediction. Or, what different times would be more useful? Can this study provide some insights?**

We decided against adding content on this for three reasons:

1. The analysis we have doesn't add much beyond what's known, which is "it depends on your time and location", e.g. late day convection in the U.S. often starting near the Rockies then progressing eastward.

2. Our AIRS-FCST goal is to target climate trends in the past. Any suggestions on when to sample wouldn't help for the past, while for the later 2020s there are planned geostationary hyperspectral IR sensors for which it doesn't make sense to suggest an overpass time.
3. To say something new will take a *lot* of investigation and very likely more than the 9-month sample we have here.

We definitely intend to investigate the diurnal cycle in AIRS-FCST with its ~20 years of data though.

**L292. The poor prediction of the temperature of the upper layers (fig. 7c) is surprising. Why?**

We think that this is consistent with a point we make later – when there's a lot of intense convection going on, ERA5 transports a lot of heat into the upper troposphere. The most obvious feature to us in Fig. 7(c) and Fig 8(b,c) is that the point "cloud" is shifted to the left near the origin, i.e. high-level cooling in ERA5-FCST but not in ERA5. To prepare readers we have added an earlier comment:

"As the convection passes overhead after 0000 UTC, it is also notable that the upper troposphere from 100—400 hPa cools substantially more in Figure 6(c) compared with Figure 6(b). The weaker cooling in ERA5 may be explained by sub-grid convection pumping heat into upper levels as the storm passes."

With this signpost we hope readers will now be able to interpret the comments later describing Figure 7. We have also added a call back to the Fig. 7 discussion:

"For middle and upper layers, the ERA5-FCST product projects larger temperature variation than occurs in ERA5, as was noted for Figure 6(b,c) after convection occurred."

In response to a comment by reviewer 2, we have added a new supplementary figure 7

"This mid- and upper-layer cooling in ERA5-FCST relative to ERA5 is indeed more common during JJA 2019 when convection peaks (Supplementary Figure 6), or during timesteps when precipitation exceeds the 99[th] percentile (Supplementary Figure 7)."

Supplementary Figure 7 is reproduced below, its caption points out how the mid- and upper-level bias described is larger in precipitating columns, which is consistent with our proposed mechanism.

[Figure]

**L321. Is this really surprising since diabatic heating tends to be balanced by adiabatic motion at grid (or large) scales? And, again, clarify what's "neglected". Another philosophical question here is that there is equal amount (50%) of unexplained variance – this raises many questions:**

**how does this limit the usefulness of the prediction, and in what situations - for example, what regions or weather systems are missed?**

We have added another sentence to explicitly call out what we think is happening:

"We expect that the unexplained variance is explained due to the contribution of the neglected diabatic processes, such as sub-grid convection, and to a lesser extent by differences between WRF versus ERA5 winds."

Combined with our other changes to your earlier comment including the explicit definition of "neglected", we think this covers the points.

**L447. Following this reasoning, shouldn't the analysis and comparison be limited to times prior to convection?**

That would be ideal, yes! In fact, one of our goals is to attempt to develop a "classifier" or similar that will tell us when convection is likely to occur. For our planned use cases, it doesn't make sense to study times prior to convection.

**Nowcasting, e.g. NUCAPS-FCST** – forecasters will have the thermodynamic fields and have the job of inferring things such as convective risk. *They do not know with certainty whether convection will happen*, so results based only on cases where convection happens would be difficult for them to use.

**Climate, e.g. AIRS-FCST** – for 2002—2013 we don't have a spatially complete, quality controlled & quantitative dataset of convective risk. For 2014—2020ish MRMS will give us that "climate quality" data, with consistent processing, sampling etc. Before MRMS we will therefore *only* have thermodynamics.

We have added text in Section 4 to try and describe, and use some Bayesian terminology to try and keep things precise and concise:

"The 2002—recent AIRS-FCST record of thermodynamics will be used with the MRMS surface radar (2014—recent) to relate the derived thermodynamics to convection. In a Bayesian sense, AIRS-FCST will provide P(thermodynamics) and to obtain our target of P(convection) we aim to derive P(convection|thermodynamics) using the combination of AIRS-FCST and MRMS. The proposed analysis is subtly different from previous work such as Kalmus et al. (2019), which studied thermodynamics in convective versus non-convective atmospheres and so reported results relevant to the inverse problem of P(thermodynamics|convection). We also emphasise that while the present study considered CAPE and CIN, this is a proof of concept that only considered a subset of potential thermodynamic properties."

---

## Author Comment (AC2)

**REVIEWER 2**

**My background is in trajectory calculations in the upper troposphere and lower stratosphere.**

**Based a method reported by Kalmus et al. (2019), this paper proposes an interesting extension to improve predictability of strong convective events from daily sun-synchronous satellite profiles by coupling with forecasts of adiabatic forward-trajectories. The paper assesses a proxy setup against ERA5 reanalysis data for the year 2019 over a part of the United States. Broader implications of the work are realised in the discussion. The results and discussion are well-written and scientifically reasonable, however I found the introduction somewhat hard to follow and some of the methods unclear. Overall, the paper is suitable for publication with ACP, however there are some specific comments that need addressing, listed below:**

Thank you for taking the time to review our paper. The perspective of a trajectory calculations expert was very helpful for us.

In particular, your review encouraged further testing of some comments we made on physical processes behind atmospheric evolution. We found support for some statements but also removed one that we could not convincingly demonstrate. The investigation was an interesting case study so we describe it in a response comment below, but do not include it in the paper since it doesn't affect any primary conclusions.

We also thank you for your very specific statements about which parts of the paper were unclear. In at least one case we had assumed knowledge of products that is not widespread, so the changes made in response to your comments should really help make our work more accessible. Importantly, we now explicitly state details that could be important context for readers to understand potential limitations.

The authors support open data & transparency and uploaded the data to the JPL Open Repository. The repository is new and the data may not be immediately downloadable, but the files will be here: https://dataverse.jpl.nasa.gov/dataset.xhtml?persistentId=doi:10.48577/jpl.EESTWM

Our point-by-point response follows. Your review text is **bold black**, our commentary is magenta (this colour!) and any quoted text that is in the main paper is in quotation marks and is "dark green".

**L64-65: Whether ERA5 is a reasonable representation of AIRS should be described. For instance, does ERA5 assimilate AIRS retrievals?**

The introduction has been extended in response to other changes, we think a discussion of ERA5's suitability fits best in the Methods Sec. 2.1 so have inserted:

"ERA5 is an ideal data source since it provides the necessary fields with horizontal resolution similar to that of AIRS and vertical resolution similar to that of our output ERA5-FCST. This work aims to evaluate the trajectory-enhancement method for adding time resolution to LEO IR products, so we are not concerned about small differences between AIRS and ERA5."

This paragraph attempts to cover the parts of ERA5 that are relevant to our goals. We will evaluate AIRS-FCST against ERA5, radiosondes and MRMS in future but our requirements here are just (i) a

sensible atmospheric state and (ii) information at fine-enough resolution to capture the structures we're looking at. Combined with the new supplementary analysis (see reviewer 1 response or later), we hope the justification is now sufficient.

**L187-188: The WRF 27km datset should be introduced appropriately. From looking around, it appears to be a series of daily forecasts and not an analysis dataset (which would prompt other questions), but the text should explain this.**

We briefly summarise this important detail, but do so earlier than where you point to in your comment. We think it makes mores sense in Sec. 2.1 where the data are introduced:

"The WRF simulations are forecasts for each UTC-defined day (Ngan and Stein, 2017)."

**L197-200: I'm a bit confused about the local enhancement metrics, mainly their names dMU_CAPE and dMU_CIN. How are they calculated for ERA5? You do not calculate the most unstable parcel for ERA5, so is dCAPE a more precise label in that case? Or are you using MU_CAPE from ERA5-FCST for the baseline in all cases? Please explain in the text somewhere, and consider the labelling convention.**

We have added discussion text along with Supplementary Figures 1—3. The associated text goes into more detail - ERA5 CAPE is for an MU parcel but uses a slightly different definition.

We changed the main text description to:

"The ERA5 CAPE is converted to an enhanced version using only the same footprints as in ERA5-FCST, and subtracting the ERA5 sample's daily median. For simplicity, we refer to this as ERA5 dMU_CAPE since ERA5 product CAPE is derived from its MU parcel (for details, see discussion associated with Supplementary Figures 1—3 and Supplementary Table 1)."

**L391 and Fig 12 caption: What is meant by time-matched? I guess you mean some sort of integration over all timesteps shown in Figs 10 and 11, but it isn't explained in the text. As part of this, I am not quite sure how you show ERA5-overpass in Figs 10 and 11 for each hour, surely you have only one set of ERA5-overpass tp values (are these the same as ERA5-FCST at t0?), I guess it is the histogram bins that are changing with hour. Again, this should be clarified in the text.**

The final paragraphs of Sec. 2.3.3 are now greatly expanded:

"Using dMU_CAPE as an example, data from all retrieval days are concatenated, resulting in N values of dMU_CAPE per UTC hour, where the N values have unique combinations of date, latitude and longitude. Thresholds in dMU_CAPE are then calculated from the percentiles of an Nx6 data array, where the x6 refers to each of the forecast UTC hours. Each location in is assigned to a percentile bin, and the associated ERA5 tp mean and frequency with which tp>4 mm hr-1 are calculated within each bin. This calculation is referred to as "matched time" and represents the performance of ERA5-FCST.

For comparison, we also calculate "overpass time" statistics in which the dMU_CAPE values are simply the N ERA5-overpass values of dMU_CAPE repeated for each UTChour. The bin edges for this sample are calculated for the same percentiles, and then the time-varying ERA5 tp statistics are calculated for this new bin assignment. In this calculation, each physical location has a single dMU_CAPE value for all six UTChours, but contributes up to six values of tp to the calculation. This represents the case of using the same nearest neighbour AIRS sounding for every forecast hour.."

Text in the Fig. 12 and 13 captions added:

"See Sec. 2.3.3 for the definition of overpass time versus matched time calculations."

**L465-467: Do you have references to support these two sentences?**

We have cited the AIRS performance test and validation report (Yue et al., 2020).

**Data and code availability: In line with the ACP data policy, the data underlying the results in this paper should be FAIR. Please provide a link to a copy of the ERA5-FCST data you generated, and if possible the analysis scripts too. Either a preliminary link for reviewers to consider, or a link to a FAIR-aligned reliable public data repository.**

We have obtained clearance to upload the ERA5-FCST data and the time-matched ERA5 fields to the JPL Open Repository (JOR). The dataset is now referenced in the data availability statement via its link, there is also an assigned DOI.

https://dataverse.jpl.nasa.gov/dataset.xhtml?persistentId=doi:10.48577/jpl.EESTWM

JPL is committed to NASA's Transform to Open Science (TOPS) and the JOR is a new part of this with the rules established only in November 2022. For now we do not believe adding the analysis code would provide enormous added value in exchange for the burden required to obtain clearance, but we fully intend to expand code availability for our work as JPL procedures evolve.

**Specific comments that do not need addressing:**
* * *
**The terms pre-convective and pro-convective are similar and risk confusing the reader, I would suggest rephrasing one or the other.**

We have replaced "pro-convective" with "conditions favourable for convection" or similar throughout.

**L240-243: Could the use of maximum unstable CAPE for ERA5-FCST be causing the higher values relative to ERA5 CAPE? If it was possible to calculate MU-CAPE for ERA5, would that also be higher? Or, how does MML CAPE for ERA5-FCST compare with ERA5 CAPE?**

This point also touches on the CAPE calculation issues discussed above, we have added the following to the text, which points to the same supplementary material that discusses CAPE calculations.

"The marginally higher mean CAPE in ERA5-overpass compared with ERA5 in Figure 2(b) may be due to differences in the ERA5 computational approach or in vertical resolution (see text associated with Supplementary Figures 1—3…)."

The supplementary analysis can't distinguish the reason because we do not have the original 137 ERA5 model level output, from which ERA5 product CAPE is calculated. We put plenty of effort into the supplementary analysis and trying to understand this point, and think we've established that our results will not be greatly affected by associated issues.

**L283-284: The argument in support of ERA5-FCST would be clearer if you could show the changes to CAPE at the different levels. Does the CAPE gained around ~800hPa outweigh losses to T-q biases near the surface?**

This is a thoughtful comment, we investigated further and removed the claim since it is not crucial to our conclusions and this example is actually rather messy. We include some analysis below that we ultimately decided was too much to put into the main paper since its inclusion would distract without greatly adding to achieving the paper's purposes.

We generally use the MU parcel, so CAPE by $P$ layers may not be informative in the Fig. 6 case. In ERA5-FCST the MU parcel moves from ~950 hPa to ~850 hPa, so for the latter period $P>850$ hPa changes are not relevant.

It's also challenging to attribute CAPE differences to any individual level or process because it doesn't decompose linearly, e.g. changes higher up can affect the (Tv-Tv,env) integral *and* the LFC/LNB integration limits.

We recalculated MU_CAPE at the Fig. 6 location with either $T(z)$ fixed at its t=0 value (MU_CAPE_dq) or $q(z)$ similarly fixed (MU_CAPE_dT). Changes relative to overpass MU_CAPE are plotted below. The magenta line is the sum of the red and blue lines, while the black line is the calculation from the ERA5-FCST profiles reported in the paper with changes in *both* T and q. Magenta follows black, so the overall time progression seems to be captured, but the decomposition is not exact since the black and magenta lines do not lie on top of each other.

The change from hours 0—6 seems to be nearly 50:50 $T$ and $q$, but the strong jump from hours 4—6 is mostly moisture driven. The $q$ changes switch the MU parcel from ~950 hPa to ~850 hPa, and the moister parcel saturates lower such that LFC drops ~70 hPa towards the surface, further boosting CAPE.

[Figure]

That story largely backs up the original phrasing. However, the direct ERA5 profiles behave *very* differently, see right panel below:

[Figure]

Clearly our argument does not apply to MU_CAPE in ERA5! The MU parcel stays nearer the surface, and therefore dries in ERA5. MU parcel drying combined with warming aloft decreases CAPE, including via raising the LFC by about 100 hPa.

This example is related to the points we touch on later (e.g. subgrid convection warming higher levels and reducing CAPE, albeit potentially non-locally) but we argue that this detailed description is just too much for the current paper. The purpose of Figs. 6 and 7 is to show the reader our process so that they can understand and interpret Fig. 8, which we consider our most-important result alongside Figs 12 & 13.

**Fig 8: The points made in the text are strong ones, but there might be a more appropriate figure. If the motivation of the eventual ERA5-FCST product is to predict hazardous convective weather, then Fig 7 is more meaningful than Fig 8, where the biases are more apparent, however it is shows a single event only. Might it be more insightful (as supplementary material) to show a version of Fig 8 restricted to periods of high-CAPE or high precipitation?**

Your suggestion re: figure 8 is a neat way to gain insight into part of what's going on, so we produced a supplementary version for grid cells where ERA5 $tp$>1.8 mm hr$^{-1}$ (~99$^{th}$ percentile). The features seem consistent with our statements throughout the paper, e.g. >50 % of variance is still captured. Secondly, we assert that ERA5-FCST gets a mid/upper level cold bias when convection happens, since convection in ERA5 causes relative warming in that product. We would expect the cold bias to get more negative during convection and that happens, the 680—440 hPa layer bias is -0.19 °C in the full sample (Fig. 8) but -1.14 °C in the precipitating columns (below).

[Figure]

We realise that there is an important subtlety in our study that was not adequately explained in the initial draft, since both you & reviewer 1 picked up on it. You both question: why are we looking at all atmospheres and not just convective cases?

Basically, we're less interested in questions "when it's raining heavily, is CAPE higher?" We're more interested in "when CAPE is higher, is it raining heavily?" This is because our product will provide CAPE (and other thermodynamic properties) rather than convective metrics. And for our planned users, we have one of two cases:

**Nowcasting, e.g. NUCAPS-FCST** – forecasters will have the thermodynamic fields and have the job of inferring things such as convective risk. *They do not know with certainty whether convection will happen*, so results based only on cases where convection happens would be difficult for them to use.

**Climate, e.g. AIRS-FCST** – for 2002—2013 we don't have a spatially complete, quality controlled & quantitative dataset of convective risk. For 2014—2020ish MRMS will give us that "climate quality" data, with consistent processing, sampling etc. Before MRMS we will therefore *only* have thermodynamics.

A new paragraph in Section 4 discusses our future goals, using some Bayesian terminology to be precise and concise:

"The 2002—recent AIRS-FCST record of thermodynamics will be used with the MRMS surface radar (2014—recent) to relate the derived thermodynamics to convection. In a Bayesian sense, AIRS-FCST will provide P(thermodynamics) and to obtain our target of P(convection) we aim to derive P(convection|thermodynamics) using the combination of AIRS-FCST and MRMS. The proposed analysis is subtly different from previous work such as Kalmus et al. (2019), which studied thermodynamics in convective versus non-convective atmospheres and so reported results relevant to the inverse problem of P(thermodynamics|convection). We also emphasise that while the present study considered CAPE and CIN, this is a proof of concept that only considered a subset of potential thermodynamic properties."

**Technical corrections:**
* * *
**L64, L78, L94, L144, L269: Tidy citation parentheses.**

Thanks for your careful reading –correction can trigger the reference plugin to force update to all references and add red "track changes" everywhere. If the paper is accepted, we will ensure that parentheses are correctly used for citations during typesetting.

**L205: Should there be a word before CAPE? low/high?**

We changed the sentence in response to the comment below.

**L205-207. This is an important sentence but is quite hard to follow. Consider rephrasing.**

We have split this into two sentences and hope it is now clearer:

"We expect that precipitation should be consistently heavier and more frequent in areas of high CAPE and/or low CIN. If high-CAPE and low-CIN conditions in ERA5-FCST are more predictive of precipitation than those conditions in ERA5-overpass, then this is good evidence of the utility of trajectory-enhancement."

**L219: A word seems to be missing from this sentence.**

We chose to delete a word instead.

**L239: Reference order for Fig 3b and 3c needs correcting.**

Done.

**Fig 8 caption: Please mention the time period being calculated over.**

Done.

**Data and code availability: Please include SHARPpy and the WRF27km dataset.**

Done.

**Supplementary figs: These refer to ERA5-AIRS-FCST, is that ERA5-FCST in the main text? Please check for consistency.**

Thanks for your careful attention, this was a good catch – it was the term used in an early draft. The figures and text have been changed to ERA5-FCST.

---

## Author Response (AR2)

Dear Editor,

We appreciate the reviewer's suggestion to perform calculations sampling ERA5 data with AIRS averaging kernels. The topic they are concerned about, namely the effective vertical resolution of the data, is indeed important.

Reprocessing the data with the AIRS averaging kernels specifically would be burdensome, since we would have to regenerate the entire ERA5-FCST dataset and all of the paper's results. The main concerns are addressed in a new sensitivity test whose results are displayed in Supplementary Tables 2—3. In main paper Section 2.1 we added:

"The AIRS L2Sup vertical layering is also far finer resolution than the "effective" vertical resolution of the retrieval, as discussed in Irion et al. (2018). In reality, the retrieval can only capture smoother changes in profiles than reported on the L2Sup layering, but we also find that our results are likely robust to the AIRS effective vertical resolution (Supplementary Tables 2—3). For FCST the finer L2Sup layering is preferred since it provides more parcels to HYSPLIT."

The only other main text changes are corrected numbering to reference later Supplementary Tables.

We think that reprocessing the entire ERA5-FCST dataset using the AIRS averaging kernels would raise issues such as how the averaging kernels do not capture the "true" information content of the AIRS spectra in new v7 retrievals, thanks to the way in which the prior is calculated. New supplementary text discusses this point and how the true effective vertical resolution may be finer than we use in our tests. Our conclusions appear robust in our strict tests and we believe we have now fully addressed all reviewer comments.

Thank you once again for your time and consideration, and to the reviewers who provided careful and thoughtful reviews.

Dr. Mark Richardson, on behalf of all authors.